

# Non-parametric learning critical behavior in Ising partition functions: PCA entropy and intrinsic dimension

**Rajat K. Panda[1,2,3]\*, Roberto Verdel[1], Alex Rodriguez[1,4],**
**Hanlin Sun[5,6], Ginestra Bianconi[5,7] and Marcello Dalmonte[1,2]**

**1** The Abdus Salam International Centre for Theoretical Physics (ICTP),
Strada Costiera 11, 34151 Trieste, Italy
**2** SISSA — International School of Advanced Studies, via Bonomea 265, 34136 Trieste, Italy
**3** INFN Sezione di Trieste, Via Valerio 2, 34127 Trieste, Italy
**4** Dipartimento di Matematica e Geoscienze, Universitá degli Studi di Trieste,
via Alfonso Valerio 12/1, 34127, Trieste, Italy
**5** School of Mathematical Sciences, Queen Mary University of London,
London, E1 4NS, United Kingdom
**6** Nordita, KTH Royal Institute of Technology and Stockholm University,
Hannes Alfvéns väg 12, SE-106 91 Stockholm, Sweden
**7** The Alan Turing Institute, 96 Euston Road, London, NW1 2DB, United Kingdom

\* rpanda@ictp.it

## Abstract

We provide and critically analyze a framework to learn critical behavior in classical partition functions through the application of non-parametric methods to data sets of thermal configurations. We illustrate our approach in phase transitions in 2D and 3D Ising models. First, we extend previous studies on the intrinsic dimension of 2D partition function data sets, by exploring the effect of volume in 3D Ising data. We find that as opposed to 2D systems for which this quantity has been successfully used in unsupervised characterizations of critical phenomena, in the 3D case its estimation is far more challenging. To circumvent this limitation, we then use the principal component analysis (PCA) entropy, a "Shannon entropy" of the normalized spectrum of the covariance matrix. We find a striking qualitative similarity to the thermodynamic entropy, which the PCA entropy approaches asymptotically. The latter allows us to extract—through a conventional finite-size scaling analysis with modest lattice sizes—the critical temperature with less than 1% error for both 2D and 3D models while being computationally efficient. The PCA entropy can readily be applied to characterize correlations and critical phenomena in a huge variety of many-body problems and suggests a (direct) link between easy-to-compute quantities and entropies.

## 1 Introduction

Classical statistical mechanics revolves around the pivotal concept of the partition function [1]. Indeed, this quantity contains *all* relevant information about a statistical system at equilibrium, and thermodynamic quantities can be obtained from it. This quantity is defined as follows:

$$Z = \sum_{\{\mathcal{S}\}} e^{-\beta H(\mathcal{S})}, \tag{1}$$

where $\beta$ is the inverse temperature, $H$ is the Hamiltonian of the system and $\{\mathcal{S}\}$ are the microscopic configurations of the system. The probability that the system is in some particular state $\mathcal{S}$ is then given by Boltzmann law, namely

$$P(\mathcal{S}) = \frac{e^{-\beta H(\mathcal{S})}}{Z}. \tag{2}$$

A full knowledge of such probabilities would then allow us to compute any expectation value of physical quantities. However, a major problem in statistical mechanics is that, for a vast majority of cases, we only know the relative but not the absolute probability. In other words, we know $e^{-\beta H(\mathcal{S})}$ but not $Z$. Powerful computational techniques such as Monte Carlo methods [2, 3] and tensor networks [4–6], provide ways to circumvent this problem and allow for an efficient evaluation of expectation values of local observables such as two-point correlators, which are crucial to characterize phase transitions. Nonetheless, a significant part of the information encoded in the partition function may be left unexplored by such traditional approaches. This is an important point to consider, in particular, for systems that feature states of matter that cannot be described by 'typical' observables.

On the other hand, these methodologies—especially, Monte Carlo simulations,— by allowing a controlled generation of large volumes of microscopic snapshots of the systems of interest, offer a fresh perspective on many-body problems as *data structure* ones [7–9]. This has brought into play powerful tools from several fields such as high-dimensional statistics, inference, and machine learning, which are being adopted more and more in the physical

sciences [10–12]. Among several methods that stem from these fields, *unsupervised learning* approaches have become prominent algorithms. Broadly speaking, such techniques aim at a characterization of the data through the understanding of underlying data relations. In condensed matter and statistical physics, such approaches have been mostly employed in the study of phase transitions and critical phenomena, including 2D [7, 9, 13–15] and 3D systems [16–18]. Additionally, there have been parallel efforts to estimate thermodynamic quantities such as the entropy—which are computationally very costly in traditional schemes—, using machine learning [19, 20] and information theoretic approaches [21], which work with reduced sampling. These previous works nonetheless call for methods that offer greater interpretability.

In this work, we put forward a theoretical approach for *learning* critical behavior in partition functions of classical spin models in an assumption-free manner. This is done by performing non-parametric statistical tests on large data sets of many-body snapshots that are sampled according to a probability distribution as in Eq. (2). We showcase our approach by studying phase transitions in two- (2D) and three-dimensional (3D) Ising models.

In the first place, we study the *intrinsic dimension* ($I_d$) [22–24] of data sets in a range of temperatures across the featured phase transitions. This concept is relevant due to the following observations: (i) the points in a data set can normally be represented as points in a high-dimensional metric space, and (ii) such points may lie on a manifold, whose (intrinsic) dimension is lower than that of the embedding space, as correlations among input variables can induce a non-trivial structure on the data. Thus, the intrinsic dimension quantifies the minimum number of variables needed to faithfully describe the data. This concept has been widely used in data science for multiple applications, for example, in the fields of molecular science [24–27] and image preprocessing [28–31]. Recently, it has been realized that structural changes in the data associated with statistical mechanical problems can reveal critical phenomena. In terms of the data manifold, this can be unveiled as a reduction of the intrinsic dimension close to the critical point [7, 32]. In particular, Mendes-Santos et al. [7], showed this for the planar Ising model and other important 2D classical lattice models. Here, we extend the aforementioned work by considering the 3D Ising model, thereby analyzing the role of the physical dimensionality on the intrinsic dimension of the data manifolds. Concretely, we use two $I_d$-estimators: (i) the two nearest neighbor (TWO-NN) method [24]—a state-of-the-art estimator based on the distribution of the ratios between second- and first-nearest neighbor distances—, and (ii) a popular projection method known as principal component analysis (PCA) [33, 34]. We find that in general, it is harder to precisely determine the phase transition of the 3D Ising model through the intrinsic dimension, compared to the 2D case. We argue that this can be regarded as a non-trivial manifestation of the higher data dimensionality concomitant to the 3D model.

Motivated by these findings, we propose a second statistical test purely based on the eigendecomposition of the covariance matrix that is done within PCA. More specifically, in analogy to Shannon's entropy [35], we define an entropy of the normalized eigenvalue spectrum of the covariance matrix. This quantity is dubbed *PCA entropy* ($S_{PCA}$). For the 2D case, we find a remarkable qualitative similarity to the exact thermodynamic entropy. In particular, $S_{PCA}$ exhibits an inflection point close to the critical temperature. This is made more explicit by considering its derivative with respect to temperature, a quantity that resembles the heat capacity, which shows a clear divergence at the transition point. From the latter, we can perform a linear finite-size scaling analysis to estimate the critical temperature with less than 1% error. Similar results hold for the 3D model, using the same amount of data as for the intrinsic dimension estimation. Hence, the PCA entropy presents itself as a versatile tool to address higher-dimensional systems where a reliable intrinsic dimension estimation may become quite challenging. We note that similar spectral entropies have been introduced in the

literature, mostly as unsupervised learning approaches for feature selection or as a measure of signal complexity. Applications range from biology [36–39] and ecology [40] to stock market dynamics [41–45] and fractals [46]. Further, the PCA entropy has very recently been introduced as a theory-agnostic measure to rank operators in quantum simulators according to their relevance (information content) [47]. While some authors work with the spectrum of a covariance matrix as in the present work, some other authors define the entropy directly with the (normalized) singular values of the data matrix (the latter approach is sometimes dubbed *SVD entropy*). Our definition is based on the spectrum of the covariance matrix, since its normalized eigenvalues give the *proportion of total variance* accounted for by the principal components, and hence, a direct measure of the relevant information contained in the principal components [10, 34].

The paper is organized as follows. In section 2, we provide a detailed description of our models and the corresponding data sets, as well as the methodology employed to create these data sets. Following that, in section 3, we focus on the intrinsic dimension estimation for the 3D Ising model. Subsections 3.1 and 3.2 delve into two different methods for estimating the intrinsic dimension, namely, the TWO-NN method and PCA, respectively. We summarize the results and shortcomings of the methods to quantitatively capture the phase transition in the considered system. In section 4, we introduce the PCA entropy, $S_{PCA}$, and show its striking qualitative resemblance to the thermodynamic entropy of Ising models. This is exploited by extracting the transition point via a finite-size analysis of its numerical derivative with respect to temperature. Finally, we draw some conclusions and discuss further potential applications of our techniques in section 5.

## 2 Models and data sets

Before exploring the different tools considered in this work, we start by defining the models and the associated data sets that we consider for our study. In this work, we investigate the 2D and 3D classical Ising model with periodic boundary conditions having nearest neighbor interaction:

$$H = -\sum_{\langle i,j \rangle} S_i S_j \,, \tag{3}$$

where $S_i = \pm 1$ are the spin degrees of freedom defined on the sites of a square and cubic lattice for 2D and 3D, respectively [48–50]. The 2D Ising model is a paradigmatic model in statistical mechanics and beyond, and it is characterized by a second-order phase transition and $\mathbb{Z}_2$ spontaneous symmetry breaking. The system goes under an order-to-disorder phase transition at the critical temperature $T_c = 2/\ln(1 + \sqrt{2}) \approx 2.269$ [48].

The exact solution of the Ising model on the simple cubic lattice is one of the long-standing open problems in rigorous statistical mechanics. The use of conformal bootstrap methods to calculate the critical exponents and critical point is still under active investigation [51, 52]. Nevertheless, multiple numerical studies, especially Monte Carlo simulations, have been done to characterize the critical properties. Similar to the case of the 2D Ising model, the 3D system features a second-order phase transition, with the critical temperature predicted at $T_c \approx 4.51$ [50, 53].

The data sets that we shall use for our subsequent analysis consist of equilibrium spin configurations of the systems introduced above. To form such data sets, we perform a stochastic sampling of the partition function of these models through Markov chain Monte Carlo (MC) simulations. Concretely, we use the Wolff cluster algorithm [54, 55], starting from the configuration with either all up spins or all down spins, chosen at random. Next, 30000 to 50000 'cluster flips' are performed for the system to equilibrate. After this, we collect $N_s = 10000$

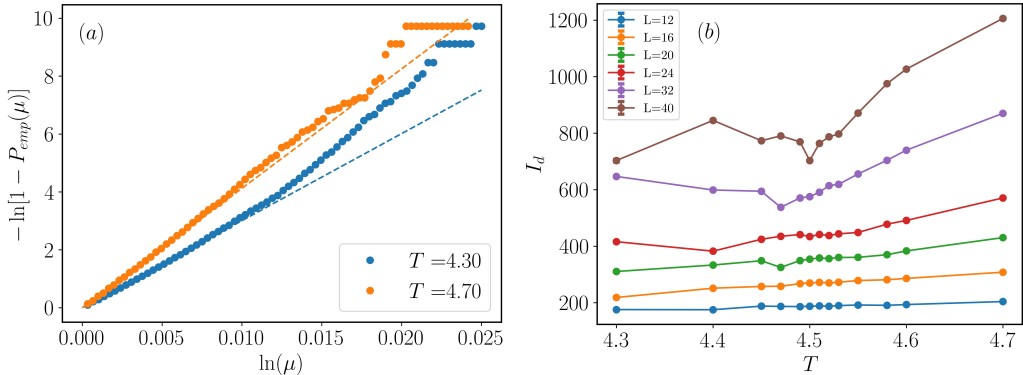

Figure 1: TWO-NN $I_d$ estimation for the 3D Ising model. (a) Empirical cumulative distribution of data for $L = 20$ and temperature 4.30 and 4.70, where the dashed line shows the linear fit used to estimate $I_d$. With the scale used in this plot, the linear fit above corresponds to Pareto distribution. (b) $I_d$ as the function of $T$ for different $L$. While we expect the $I_d$ to increase at high temperatures, and to drop as $T \to 0$, close to the transition point, this quantity features a local minimum, which becomes more apparent as the system size is increased.

state configurations, $\{\vec{x}^i \equiv (S_1^i, S_2^i, \ldots, S_N^i)\}_{i=1}^{N_s}$, where $S_n^i$ is the spin variable at site $n$ in the $i$-th realization, and $N = L^2$ for $D = 2$ or $N = L^3$ for $D = 3$, with $L$ being the linear size of the system. Importantly, the collected configurations are separated by a number of cluster flips in the range of 1000 to 1500, so as to have as little correlation among them as possible (for a detailed discussion on decorrelation of sampled state configurations and autocorrelation times see Appendix A).

For each temperature, we perform five independent MC simulations as described above, leading to an overall number of sampled configurations of 50000. This constitutes the total number of points in the data set at a given temperature. To perform statistics, we then use a subsampling algorithm [56, 57], wherein $N_b$ 'batches' of data are formed by selecting, for each of them, $N_r = 10000$ configurations at random but without repetitions from the whole ensemble of 50000 sampled configurations. Each batch of data is then represented as a matrix with $N_r$ rows (number of realizations) and $N$ columns (number of degrees of freedom), that is, $\mathbf{X} = \{\vec{x}^1, \vec{x}^2, \ldots, \vec{x}^{N_r}\}$. For more details on the subsampling technique, see Appendix B.

## 3 Intrinsic dimension

High-dimensional data sets usually have hidden internal structures which essentially live on low-dimensional manifolds. Such manifolds can then be described—without losing relevant information—by a smaller number of features than the embedding dimension. The reduced number of variables needed to describe the data is known as *intrinsic dimension*, $I_d$ [22, 23]. This key observation is the reason for the great success of dimensional reduction algorithms. However, estimating the $I_d$ of high-dimensional data sets is a problem that is far from trivial, since the corresponding data manifolds might be highly curved and twisted. Hence, this is an active field of research, with however some recent methodologies that have been shown to be able to mitigate the effects of curvature and inhomogeneities [24].

On the other hand, recent studies have shown the versatility and potential of the $I_d$ as an unsupervised learning scheme to study critical phenomena in a variety of classical [7] and quantum [32] statistical mechanical models. Nonetheless, thus far, such efforts have been

only carried out for low-dimensional systems. A systematic study of how volume can affect the estimation of the $I_d$ of data sets associated to such many-body problems, hence, remains an open question. In our work, we take a first step along this direction by systematically investigating the intrinsic dimension in the 3D Ising model. Specifically, we use two different methods to estimate the $I_d$ of data sets, namely, TWO-NN [24] and PCA [33].

## 3.1 TWO-NN method

Although there are multiple ways to calculate the $I_d$, the two-NN method has recently gained popularity for its versatility in dealing with very high-dimensional data sets. This $I_d$-estimator only relies on the statistics of distances to each point's first two nearest neighbors. The method is rooted in computing the distribution function of neighborhood distances, which are functions of $I_d$. For every point $\vec{x}_i$, the first two nearest neighbor distances $r_1(\vec{x}_i)$ and $r_2(\vec{x}_i)$ and the ratio $\mu_i = r_2(\vec{x}_i)/r_1(\vec{x}_i)$ are calculated. Under the condition that the data set is locally uniform in the range of next-nearest neighbors, it has been shown in Ref. [24] that the distribution function of $\mu$ is given by

$$f(\mu) = I_d \mu^{-I_d - 1} . \tag{4}$$

From the cumulative distribution (CDF) of $f(\mu)$, denoted $P(\mu)$, we then obtain

$$I_d = -\frac{\ln[1 - P(\mu)]}{\ln(\mu)} . \tag{5}$$

In practice, one can use the empirical CDF, $P_{emp}(\mu)$, together with Eq. (5) to estimate the $I_d$ by a linear fit of the points $\{(\ln(\mu), -\ln[1 - P_{emp}(\mu)])\}$, passing through the origin as illustrated in Fig. 1(a).

In Fig. 1(b), we plot the estimated values of $I_d$ as a function of temperature, for varying system size $L$, for the 3D Ising model. Though for small system sizes, there is no noticeable signal in the behavior of the $I_d$, as the system size is increased we observed that (i) at high temperatures the $I_d$ monotonically increases as expected (since high-temperature Ising snapshots correspond to disordered (random) spin configurations), and (ii) most remarkable, around the transition point, the $I_d$ features a local minimum. As understood for 2D classical spin systems featuring continuous phase transitions, at the transition point the system becomes parametrically simpler due to universality, which in turn simplifies the concomitant data structure [7]. However, because of the higher dimensionality of the problem, unlike for the 2D Ising model, here we observe that the signal is weaker (for the accessed system sizes), making it significantly more challenging to get a reliable quantitative characterization of the phase transition through the $I_d$.

## 3.2 PCA-based $I_d$ estimation

We now try a different approach to estimate the $I_d$ of 3D Ising partition functions data sets. Specifically, we use the popular non-parametric technique known as PCA. The main idea behind PCA is that the essential information within a data set is contained in the variability of the data. Hence, one aims at finding the directions along which the data exhibit the highest variance. This can be accomplished by means of a linear transformation of the set of coordinates [33]. The procedure to find such high-variance directions can be approached in different ways, for example, by diagonalizing the covariance matrix, or equivalently, by performing a singular value decomposition (SVD) of the data matrix [33, 34]. These approaches are briefly explained below. Each data set is represented by a rectangular matrix $\mathbf{X}[N_r, N]$, having the Monte Carlo snapshots as its rows, with $N_r$ being the sample size. For convenience, we subtract the mean of each column from the entries of the columns to obtain the "centered data

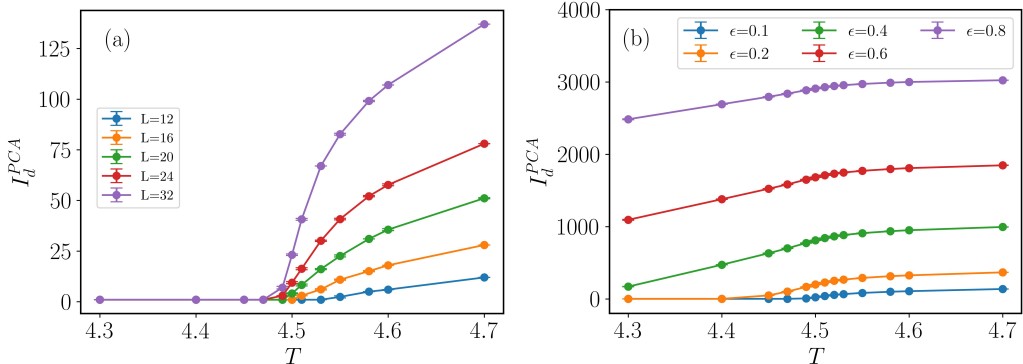

Figure 2: PCA-based $I_d$ estimation for 3D Ising model. (a) $I_d^{PCA}$ as a function of $T$ for different system sizes with cutoff $\epsilon = 0.10$. For such an *ad hoc* cutoff, $I_d^{PCA}$ abruptly drops to 1 below $T_c \approx 4.51$, while it rises above the transition. (b) $I_d^{PCA}$ for $L = 32$, with varying cutoff $\epsilon$ [see Eq. (9)]. For sufficiently large values of $\epsilon$, $I_d^{PCA}$ does not drop to 1 just below the transition point. However, a signature of the transition can be observed as a visible change in the slope around $T_c \approx 4.51$.

matrix", $\mathbf{X}^*$ [9]. In this case, the sample covariance matrix can be estimated as [33,34]

$$\Sigma = \frac{1}{N_r - 1}\mathbf{X}^{*T}\mathbf{X}^*, \tag{6}$$

which is a $N \times N$ symmetric matrix ($\mathbf{X}^{*T}$ is the transpose of $\mathbf{X}^*$). It can be shown that the principal axis and their variance are defined, respectively, by the eigenvectors and eigenvalues of this matrix, which are obtained by solving the eigenvalue problem

$$\Sigma\vec{w}_n = \lambda_n\vec{w}_n. \tag{7}$$

In practice, it is convenient to determine these quantities through an SVD of $\mathbf{X}^*$. In effect, one can readily show that the eigenvalues of $\Sigma$ are proportional to the squared singular values of $\mathbf{X}^*$. Here we perform a full SVD on the matrix $\mathbf{X}^*$ using the package *scikit-learn* [58], which gives us $\lambda_1 \geq \lambda_2 \geq \cdots \geq \lambda_k \geq 0$, where $k$ is the rank of $\mathbf{X}^*$, that is, $k \leq min(N_r, N)$. We then define the normalized eigenvalues, which is a standard measure to quantify the proportion of the total variance that is accounted for by the corresponding principal component. Namely,

$$\tilde{\lambda}_n = \frac{\lambda_n}{\sum_{m=1}^{k} \lambda_m}. \tag{8}$$

The $I_D^{PCA}$ can then be defined by choosing an *ad hoc* cutoff parameter $\epsilon$ for the integrated normalized spectrum of the covariance matrix [33]

$$\sum_{n=1}^{I_d^{PCA}} \tilde{\lambda}_n \approx \epsilon. \tag{9}$$

As discussed in recent works (see, in particular, Ref. [7]), the PCA-based $I_d$ estimation differs from the TWO-NN one, in that the former can be regarded as a *global* estimator, while the latter is a *local* one. The implication of this fact is that rather than featuring a local minimum around the transition point, $I_d^{PCA}$ drastically drops to 1 below the transition point [7]. We recover such behavior in the 3D case, too, specifically, for a value of the cutoff parameter of $\epsilon \sim 0.1$; see Fig. 2(a). We note that such a value is much smaller than the reported value in the case of 2D

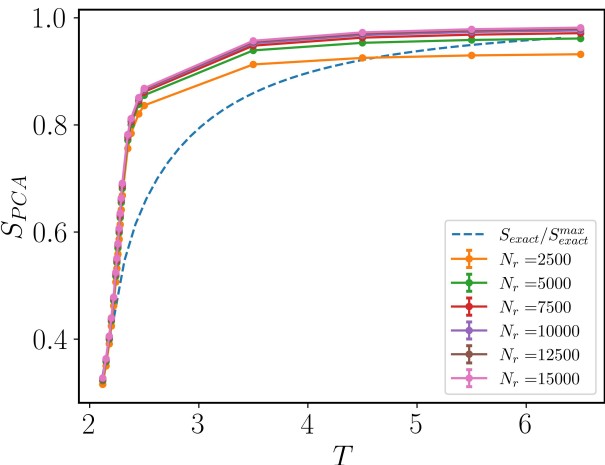

Figure 3: Comparison of $S_{PCA}$, for different sample sizes $N_r$, with the exact thermodynamic entropy per spin of the 2D Ising model with $L = 48$ as a function of temperature. Both entropies have been normalized such that their maximum possible value is 1 [see Eq. (10)].

Ising [7]. We ascribe this as the clear signature of a non-trivial volume effect in the 3D case, which suppresses the dominance of the biggest contributing explained variance $\tilde{\lambda}_1$. As we can observe in Fig. 2(b), for different values of the cutoff parameter $\epsilon$, $I_d^{pca}$ can vary significantly and require substantial fine-tuning to find the working window for the cutoff. Nevertheless, we note that even in those cases, a signature of the transition is still clearly visible through the form of a change in the slope of $I_d^{PCA}$.

In summary, the intrinsic dimension obtained via PCA can indeed host signatures of a phase transition, however, their visibility—and in fact, even their nature—is very sensitive to the choice of the cutoff parameter, signaling a degree of arbitrariness, and also making it challenging to obtain controlled estimates for the case of the 3D Ising model.

## 4 PCA entropy

In order to circumvent the aforementioned difficulties in the unsupervised characterization of phase transitions in higher dimensional systems using $I_d$-based approaches, we now consider a complementary measure of data set complexity, namely, the *PCA entropy*, $S_{PCA}$. This quantity—and the closely related *SVD* entropy—has recently been employed in unsupervised schemes for feature selection in biology [37–39], to quantify the complexity of ecological networks [40] and financial time series [41–45], and even in the characterization of the dimension of fractals [46]. Further, very recently, this quantity has been employed as an unbiased metric to rank operators in quantum simulators based on their relevance (information content) [47]. It is one of the primary goals of this work to show that $S_{PCA}$ can readily be used to characterize correlations and critical phenomena in other many-body problems as well. In the particular context of this work, we shall see that this quantity is less sensitive to volume effects, as opposed to the $I_d$ estimators discussed above. At the same time, as we will illustrate for the Ising models under consideration, this quantity bears a remarkable qualitative resemblance with the thermodynamic entropy. Importantly, the calculation of the PCA entropy is computationally very amenable.

The starting point to define the PCA entropy is the eigendecomposition of the sample covariance matrix. Specifically, by noticing that the normalized eigenvalues $\tilde{\lambda}_n$ in Eq. (8) satisfy

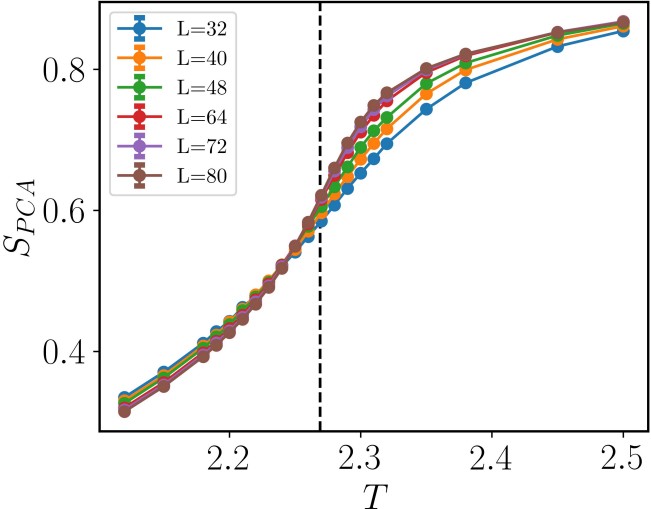

Figure 4: $S_{PCA}$ as a function of temperature for different system sizes $L = 32 - 80$, for 2D Ising model. These plots exhibit a clear crossing point in the vicinity of the transition point, suggestive of a finite-size scaling analysis.

that (i) $\tilde{\lambda}_n \geq 0$ for all $n$ (as they are proportional to the squared singular values of $\mathbf{X}^*$), and (ii) $\sum_n \tilde{\lambda}_n = 1$ (by construction), we can follow Shannon's entropy formula [35] to define

$$S_{\text{PCA}} := -\frac{1}{\ln(k)} \sum_{n=1}^{k} \tilde{\lambda}_n \ln(\tilde{\lambda}_n). \tag{10}$$

In general, the PCA entropy in Eq. (10) can be used as a measure of the correlations among the input variables in the analyzed data set. Indeed, note that for an extremely 'correlated' data set, which under PCA can be fully described by a single principal component (i.e., $\tilde{\lambda}_1 \sim 1$, $\tilde{\lambda}_n \sim 0$, for $n \geq 2$), we get $S_{\text{PCA}} = 0$. Instead, for a fully 'uncorrelated' data set (e.g., a collection of independent random variables), for which $\tilde{\lambda}_n = 1/k$ for all $n$, we have $S_{\text{PCA}} = 1$. Note that with the definition in Eq. (10), the maximum value that $S_{PCA}$ can take is precisely 1.

Physically it is then clear that in the limits of $T \to 0$ and $T \to \infty$, for which the data sets are very 'ordered' and 'random-like', respectively, the behavior of $S_{PCA}$ should, at least qualitatively, correspond to that of the thermodynamic entropy, that is, we expect $S_{PCA}$ to vanish as $T \to 0$ and $S_{PCA} \sim 1$ as $T \to \infty$. That is exactly what we observe in Fig. 3, where we plot $S_{PCA}$ for varying number of sample sizes ($N_r$), in the case of the 2D Ising model with $L = 48$. Furthermore, we compare those curves with the exact thermodynamic entropy per spin, which is computed using the explicit solution for finite square lattices with periodic boundary conditions (see, for instance, Refs. [20, 60, 61]). Note that the latter entropy is also normalized by its maximum possible value, in order to facilitate a direct comparison. This comparison suggests that $S_{PCA}$ should asymptotically coincide with the thermodynamic entropy as $T \to \infty$ and $N_r \to \infty$. Apart from this limit, it is still quite remarkable the qualitative similarity between these two entropies, as already anticipated, even more so, as this is achieved even with reduced sampling, for example, $N_r = 2500$ in Fig. 3, which requires a very modest computational overhead.

In Fig. 4, we plot $S_{PCA}$ for the 2D Ising model for different system sizes in a reduced range of temperatures around the transition point. In these and further calculations, we have fixed $N_r = 10000$. We note that $S_{PCA}$ features a flex close to the transition point, which is immediately highlighted by the crossing of the curves when varying the system size $L$. This suggests a finite-size scaling analysis. To perform such an analysis in a more accurate way

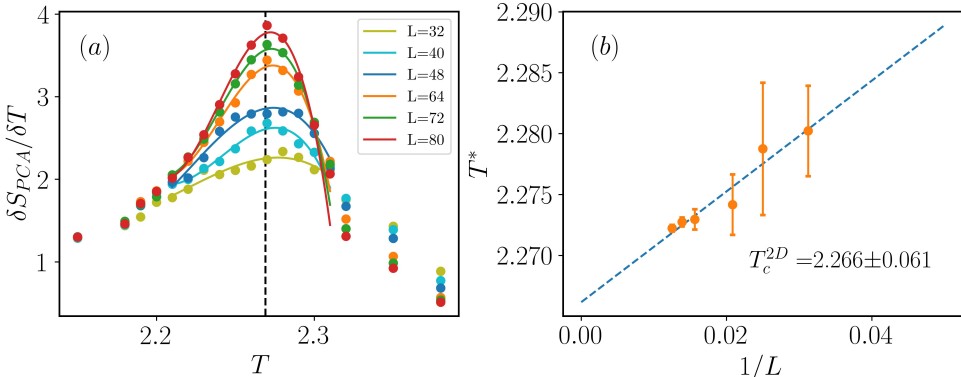

Figure 5: (a) Plot of $\delta S_{PCA}/\delta T$ as a function of temperature for the 2D Ising model. The location of the flex in $S_{PCA}$ is revealed by the peak in its derivative, occurring at $T^*(L)$. Solid lines show a smoothing curve of the data obtained via a standard smoothing spline function. (b) Linear finite-size scaling of the temperature where we get the maxima $T^*(L)$. This linear fit yields $T_c^{2D} = 2.266 \pm 0.061$.

and allow for quantitative predictions, we compute the numerical derivative of $S_{PCA}$, which we approximate here by its symmetric difference quotient:

$$\frac{\delta S_{PCA}}{\delta T} := \frac{S_{PCA}(T + \Delta T) - S_{PCA}(T - \Delta T)}{2\Delta T} . \tag{11}$$

This is shown in Fig. 5(a) for the 2D Ising model. We use a smooth spline approximation (using the function *splrep* from the package *scipy* [62]), to smooth out the curves and track the temperature at which they feature a local maximum, $T^*(L)$. The temperature window in which we perform the smoothing spline is $T \in [2.2, 2.31]$; solid lines in Fig. 5(a). This allows us to carry out a linear finite-size scaling analysis as shown in Fig. 5(b), which leads to an estimated critical temperature $T_c^{2D} = 2.266 \pm 0.061$, in excellent agreement with the exact value. In Fig. 5(b), we observe larger error bars for smaller system sizes. This is a consequence of the numerical derivative not having a prominent peak (and, in addition, the interpolation is more affected by fluctuations in the sampling when compared to larger lattices). For larger system sizes the peak in the derivatives is more pronounced and the calculation of $T^*(L)$ is less noisy; hence, we observe the sharp decrease in error bars. The error bars have been computed using the subsampling procedure explained in Sec. 2 and Appendix B, averaging over 10 subsamples of data, each containing $N_r = 10000$ data points. That is, for each batch of data we get a smooth spline approximation, and extract the location $T_i^*$ of the corresponding local maxima. We then compute the mean $\overline{T^*}$ and the subsampling error.

The corresponding results for the 3D Ising model are shown in Figs. 6 and 7. First, we note that, as opposed to the 2D case, the curves of $S_{PCA}$ do not clearly cross as we vary $L$. This is most likely due to the fact that since we have fixed $N_r = 10000$, there will be a different normalization factor, $\ln(k)$, in the definition in Eq. (10) depending on whether $L^3$ is smaller of bigger than $N_r$. Indeed, if $N = L^3 < N_r$, then $k = N$, otherwise $k = N_r$. (For all the values of $L$ considered in Fig. 4, it is always that case that $N = L^2 < N_r$, and hence, we always normalize the entropy by $\ln(N)$. Imposing a similar constraint in the 3D case would yield a bigger computational overhead or limit us in the system sizes that we can consider.) Yet, a similar analysis using the derivative of $S_{PCA}$, shown in Fig. 7(a), where we have used a temperature window $T \in [4.4 - 4.53]$ for the smoothing spline; solid lines in Fig. 7(a), allows us to perform a similar linear finite-size scaling analysis, shown in Fig. 7(b), yielding an estimated critical temperature $T_c^{3D} = 4.518 \pm 0.070$, once again in very good agreement with the reported value in the literature.

Finally, we note that the smoothing splines shown in Figs. 5(a) and 7(a), were done using a smoothing condition parameter $s$ [62], so that $\sum_i (g_i - y_i)^2 \leq s$, where $g(x)$ is the smoothed interpolation of $(x, y)$. In practice, we found that setting $s$ to less than 1% of the maximum of $y$ gives stable results, and concretely, we set $s = 0.05$ and $s = 0.005$ for 2D and 3D Ising model respectively.

# 5 Conclusions and outlook

In summary, we have introduced a theoretical framework to learn critical behavior in partition functions of classical systems using non-parametric unsupervised approaches. We have showcased our methods by studying phase transitions in classical Ising models in 2D and 3D rectangular lattices, harvesting thermal configurations from MC simulations. In the first place, we have unveiled the role of volume in the estimation of the intrinsic dimension of data sets of thermal MC configurations. The intrinsic dimension is widely used in machine learning and has recently been applied in unsupervised studies of critical phenomena in 2D classical systems. We explored this property for the first time in 3D systems. We found that, while it is still possible to detect the transition point with reasonable accuracy through changes in the behavior of this quantity as a function of temperature, in general, its estimation becomes much more challenging than in the 2D case. The latter holds when using both local and global estimators such as the TWO-NN method and PCA, respectively. Further, this observation is very likely a direct manifestation of what in data science is known as the *curse of dimensionality* [59]. In the quest to overcome this difficulty, we have then introduced the concept of *PCA entropy*—a "Shannon entropy" of the normalized spectrum of the covariance matrix. This and related spectral entropies are widely used in unsupervised approaches for feature selection tasks as well as a measure of signal complexity. Here, we have applied this quantity for the first time to data sets of statistical mechanics systems and found a striking qualitative similarity with the thermodynamic entropy of the Ising model, exhibiting in particular a flex around the transition point, both for the 2D and 3D cases. This allows for a very accurate estimation of the critical temperature (with less than 1% error) by a conventional finite-size scaling analysis. Further, we have argued how the PCA entropy can asymptotically recover the thermodynamic

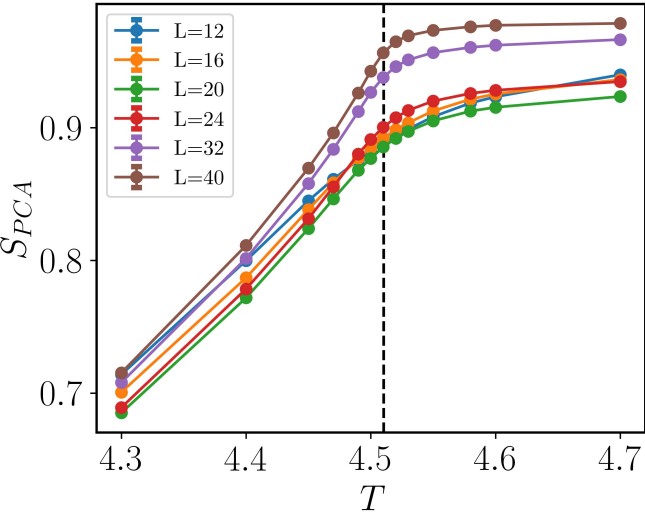

Figure 6: $S_{PCA}$ as a function of temperature for different system sizes $L = 12 - 40$, for 3D Ising model.

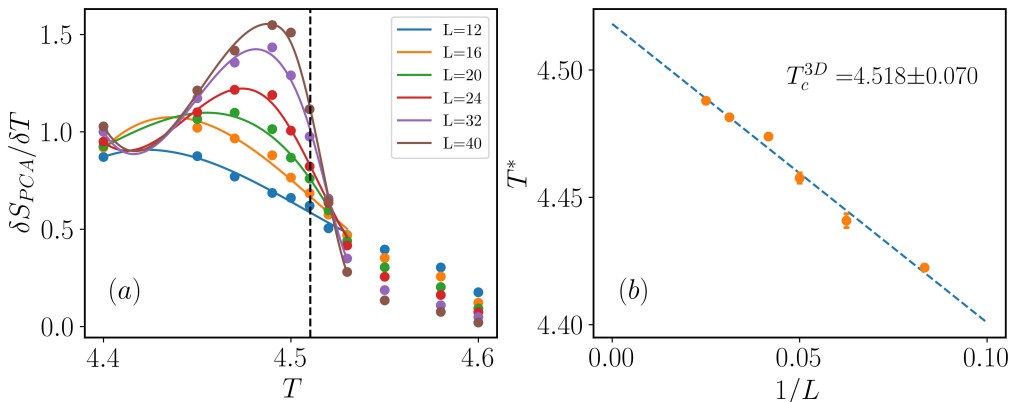

Figure 7: (a) Plot of $\delta S_{PCA}/\delta T$ as a function of temperature for the 3D Ising model. The location of the flex in $S_{PCA}$ is revealed by the peak in its derivative, occurring at $T^*(L)$. Solid lines show a smoothing curve of the data obtained via a standard smoothing spline function. (b) Linear finite-size scaling of the temperature where we get the maxima $T^*(L)$. This linear fit yields $T_c^{3D} = 4.518 \pm 0.070$.

entropy while being computationally efficient and interpretable—as opposed to other machine learning approaches—due to its own definition.

Several interesting questions remain as directions for future research. In particular, it would be very interesting to see the scope of the PCA entropy in the study of different kinds of phase transitions such as Berezinskii-Kosterlitz-Thouless (BKT) transitions. In this respect, analyzing whether and how the PCA entropy is sensible to the effects of topology is a question well deserving of attention. Besides, characterizing the limitations of the intrinsic dimension due to volume effects in different systems is another critical question to be explored. Additionally, our methods can readily be applied to learn path integrals of quantum statistical systems, thereby complementing and extending previous theoretical works [32]. Finally, the analysis of experimental data sets associated with many-body problems is also immediately within reach, as already exemplified in recent related works [47,63]. Along a separate route, it is essential to mention that the dimensional analysis performed here indicates that manifolds describing partition functions are in fact very rich and correlated: a very promising route to unfold such correlations is provided by network theory—that we are illustrating, in the context of Ising partition functions, in a parallel work [64].

## Acknowledgments

We are grateful to S. Acevedo, A. Laio, B. Lucini, T. Mendes-Santos, S. Pedrielli, and V. Vitale for discussions and feedback on this and related works.

**Funding information**    This work was partly supported by the MIUR Programme FARE (MEPH), by QUANTERA DYNAMITE PCI2022-132919, and by the PNRR MUR project PE0000023-NQSTI. MD is also supported in part by the PRIN programme (project CoQuS).

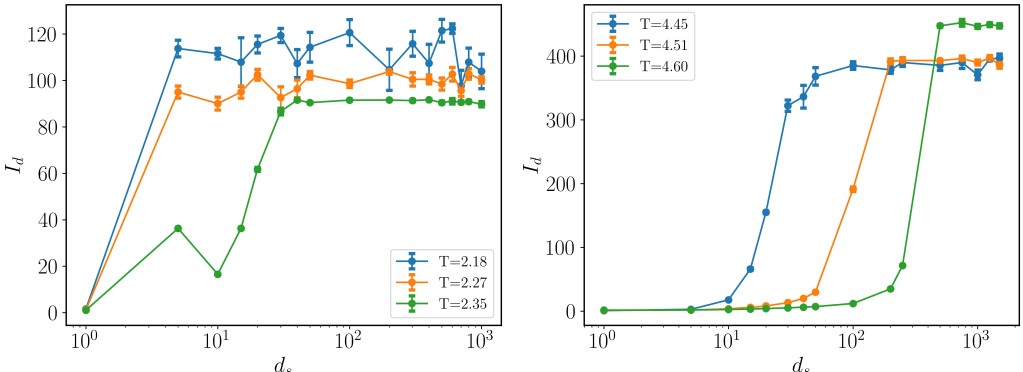

Figure 8: $I_d$ as a function of the sampling interval $d_s$, for 2D and 3D Ising partition function data sets, respectively, at different temperatures. After some transient behavior, the $I_d$ saturates at some given value and does not change further. This insensitivity with respect to the sampling interval signals the point after which configurations sampled during the MC simulations are essentially uncorrelated from each other. This defines the *decorrelation time* $d_s^\star$ (see main text).

## A  Analysis of the decorrelation of state configurations via $I_d$ and $S_{PCA}$

In this appendix, we elaborate on how we minimize the correlation between the configuration extracted from the Monte Carlo simulation. In order to make sure that we have attained the desired data set with decorrelated configurations, we study $I_d$ and $S_{PCA}$ as the function of sampling interval $d_s$, the number of Wolff's cluster flips between two consecutive configurations saved. For all the calculations below we have $N_r = 5000$ with the configurations taken from the same Monte Carlo simulation and averaged over 5 realizations. The system sizes are fixed, $L = 48$ for 2D and $L = 24$ for 3D Ising.

In Fig. 8, we can observe that after an initial increase in $I_d$ with $d_s$, the $I_d$ value saturates and stabilizes within error bars for increments of $d_s$. The point after which the $I_d$ saturates indicates the minimum value of $d_s$ required to build the uncorrelated data set. We call such a value *decorrelation time* and denoted $d_s^\star$. This decorrelation time increases with temperature: for 2D at $T = 2.27$, $d_s^\star \simeq 10$ seems to be enough to decorrelate the configurations, but for $T = 2.35$ we need $d_s^\star \simeq 40$. We observe a similar trend for the 3D case with increasing temperature requiring higher times: at $T = 4.60$, we get $d_s^\star \simeq 500$. In practice, however, we set a final sampling interval to be at least two or three times $d_s^\star$, which for the latter case, for example, corresponds to $1000 - 1500$ cluster flips in between sampled state configurations.

In Fig. 9 what we observe for $S_{PCA}$ complements the findings from $I_d$, with $S_{PCA}$ rising and eventually saturating at some given value as $d_s$ is increased. We note that, in general, the values of $d_s^\star$ that can be read out from the latter plots are compatible with those estimated using the $I_d$, both in 2D and 3D.

We note that the decorrelation time $d_s^\star$ is an intrinsic property of the specific algorithm utilized to carry out the MC sampling. This is analogous to the *autocorrelation time*, which is a paramount quantity to analyze in any MC simulation. Indeed, the key difficulty in (dynamic) MC is that the successive states in the underlying Markov chain are correlated, naturally increasing the error of estimates [65]. For some given observable, for example, the magneti-

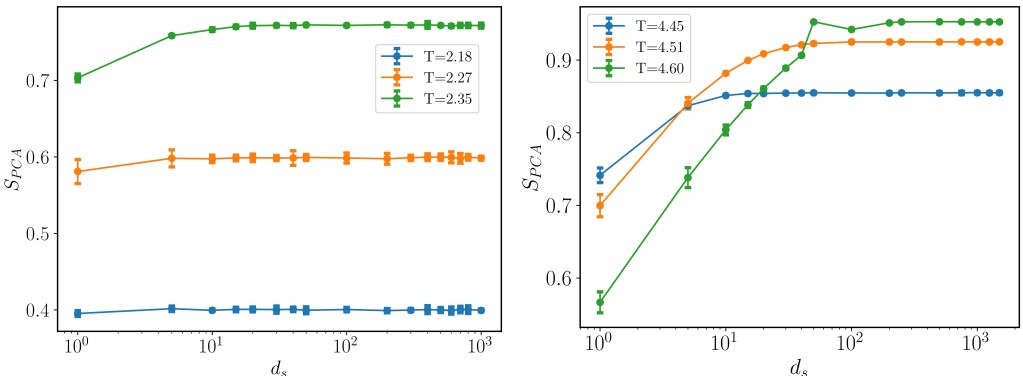

Figure 9: $S_{PCA}$ as a function of the sampling interval $d_s$, for 2D and 3D Ising partition function data sets, respectively, at different temperatures. We observe compatible values of the $d_s^\star$, with those estimated from the $I_d$ analysis; c.f. Fig. 8.

zation $M$, the autocorrelation function $C(t)$ as a function of the MC time $t$ is given by

$$C(t) = \frac{\left\langle M_j M_{j+t} \right\rangle - \langle M \rangle^2}{\langle M^2 \rangle - \langle M \rangle^2}, \tag{A.1}$$

with $j$ denoting some reference time, which we can choose arbitrarily since at equilibrium time translational invariance holds. In the above definition, we use

$$\langle M^\alpha \rangle = \frac{1}{N_t} \sum_{i=1}^{N_t} M_i^\alpha, \qquad \left\langle M_j M_{j+t} \right\rangle = \frac{1}{N_t - t} \sum_{i=1}^{N_t - t} M_i M_{i+t}, \tag{A.2}$$

where $N_t$ is the total number of MC steps.

For well-formulated algorithms, it is typically expected that the autocorrelation function introduced above will decay exponentially with $t$, that is,

$$C(t) \simeq \exp(\tau/t), \tag{A.3}$$

where $\tau$ is the autocorrelation time of the observable in the given algorithm. To be more precise, this time is called the exponential autocorrelation time $\tau_{exp}$. A second autocorrelation time is so-called *integrated autocorrelation time* (IAT), $\tau_{int}$, which determines the statistical errors in the MC estimates of observables [65]. The latter can be estimated as follows

$$\tau_{int}(W) = \frac{1}{2} \sum_{t=1}^{W-1} C(t) + R(W), \tag{A.4}$$

with

$$R(W) = \frac{C(W)}{1 - \frac{C(W)}{C(W-1)}}, \tag{A.5}$$

that shall converge fast for $W \gg 1$.

We computed the IAT for the temperatures above for the 2D and 3D systems, using 10000 successive configurations after the equilibration. We found $\tau_{int} \simeq 33, 40,$ and 48 for $T = 2.18,$ 2.27, and 2.35 respectively in the 2D case. For the 3D case we get $\tau_{int} \simeq 46, 51,$ and 54 for $T = 4.45, 4.51,$ and 4.60 respectively.

Whether or not the autocorrelation time of observables and the decorrelation time estimated via the $I_d$ and $S_{PCA}$ analyses can be related to each other is a question well deserving a more in-depth exploration, which however we leave for future research. Nevertheless, we should mention that the used decorrelation times, as defined above, are a crucial piece of information to ensure the reproducibility of the results discussed in this work.

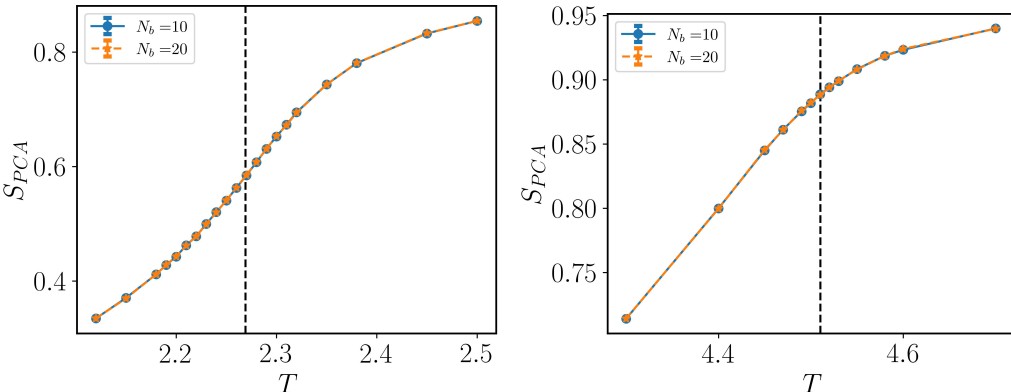

Figure 10: $S_{PCA}$ as a function of temperature for different values of $N_b$, for 2D and 3D Ising partition function data sets. The system size for 2D is $L = 32$ and for 3D $L = 12$.

## B  Subsampling

In this appendix, we describe the subsampling algorithm used to perform statistics on the collected data and establish the corresponding error bars.

Given a data set with a total number of points $N_T$: $\mathbf{X} \equiv \{\vec{x}^1, \ldots, \vec{x}^{N_T}\}$, we repeatedly compute a quantity of interest $\phi$ on $N_b$ 'batches' (subsamples) of data, which are obtained by randomly drawing samples of size $N_r$ *without* replacement from the finite population $\{\vec{x}^1, \ldots, \vec{x}^{N_T}\}$. We denote such estimates as $\phi(\mathbf{X}_\beta)$, with $\beta = 1, \ldots, N_b$. From these estimates, we can compute the sample mean:

$$\overline{\phi} = \frac{1}{N_b} \sum_{\beta=1}^{N_b} \phi(\mathbf{X}_\beta). \tag{B.1}$$

The associated standard error can be estimated as [56, 57]

$$SE \approx \sqrt{\frac{N_r}{N_T - N_r}} \times \sqrt{\frac{1}{N_b} \sum_\beta (\phi(\mathbf{X}_\beta) - \overline{\phi})^2}. \tag{B.2}$$

This formula is known as the (stochastic) delete-$d$ Jackknife standard error estimator (with $d = N_T - N_r$), which is usually employed within subsampling schemes [56]. We note that this method is also related to the bootstrap method [57], with the main difference that samples are drawn without replacement. The latter fact is crucial, for example, when estimating the $I_d$ through the TWO-NN algorithm, which works under the assumption of no repetitions in the considered data points (if repetitions occur, different estimators based on discrete distances can be employed [66]).

Finally, it can be shown that under adequate conditions the distribution of $\phi(\mathbf{X}_\beta)$ will converge to the sampling distribution of $\phi$. In particular it is required that $N_r \to \infty$ as $N_T \to \infty$, but with $N_r/N_T \to 0$. In practice, the choice of the parameters above is data-dependent. Here, we have $N_T = 50000$, and found consistent results with the choices $N_b = 10$ and $N_r = 10000$ (unless otherwise specified), as mentioned in the main text.

In Fig. 10, we check the effects on $S_{PCA}$ while changing $N_b$ for some fixed system size. We find negligible change in $S_{PCA}$ value for changing $N_b = 10$ to $N_b = 20$ in the case of both 2D and 3D Ising partition function data sets.

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
