# Peer review of "Non-parametric learning critical behavior in Ising partition functions: PCA entropy and intrinsic dimension"

_SciPost Physics Core, doi:SciPost Phys. Core 6, 086 (2023)_

## Round 2 · Referee Report · Biagio Lucini (Referee 1) · 2023-10-17

Strengths

(1) The paper provides a robust analysis of phase transitions using two different unsupervised machine learning methods.
(2) The paper contrasts strengths and weaknesses of the two approaches in a sound way.
(3) While the paper uses the Ising model as a test system, conclusions seem generalisable, with the results possibly opening new avenues of investigation in Statistical Mechanics and Quantum Field Theory at finite temperature.

Weaknesses

In my opinion, the paper does not have any weaknesses. There are some technical points that I would like the authors to clarify, but I consider those addressable at this stage. Details are in the report below.

Report

Machine learning methods are gaining adoption also in fundamental physics as tools for getting insides on physical systems. In this work, the Authors explore two unsupervised methods, the intrinsic dimensionality, and the PCA entropy, as tools to gain quantitative information on phase transitions. The intrinsic dimensionality approach is applied to the 3D Ising model, following previous promising explorations in 2D. The authors find that as a tool this method is rather inefficient in the 3D case, suggesting the curse of dimensionality as an explanation. The second method, the PCA entropy, is more promising and provides very good results. The method is physically insightful and generalizable to other systems. As far as I know, this proposal is original. Therefore, in terms of originality and interest, I believe that the paper meets the mandatory criteria of Scipost Physics Core. In addition, in my opinion, the paper also meet the general criteria (e.g., in terms of clear structure, sound conclusions, very good English, clarity etc.).

However, before I can recommend the work for publication, I would like the authors to consider the changes I suggest below.

Requested changes

(1) The value Nb = 10 seems rather small for the central limit theorem to apply, which is a general requirement of resampling methods. The authors should comment on the choice of Nb in appendix B, providing an example of what happens for a larger value (e.g., Nb = 20);
(2) The authors should explain the size of the errors in Fig. 5b, which appears to be of very different magnitudes, and sharply decreasing for larger lattices.
(3) It is not clear to me whether the interpolation error for the computation of the PCA entropy (e.g., Fig. 5a) has been accounted for. This could be done, e.g., by double resampling, but possibly there are simpler ways to estimate it. The authors should provide an estimate of this error with a short explanatory discussion.

  • validity: high
  • significance: high
  • originality: top
  • clarity: top
  • formatting: perfect
  • grammar: perfect

Author:  Rajat Panda  on 2023-11-13  [id 4109]

(in reply to Report 1 by Biagio Lucini on 2023-10-17)

We thank the Referee for their time in critically assessing our manuscript and for the high evaluation of our work.

Below we address the requested changes.

(1) We thank the Referee for raising this point. In Appendix B, we have added a discussion on the effect of changing $N_b$ on $S_{PCA}$. In both the 2D and 3D cases, by changing the value of the number of batches $N_b$, from $10$ to $20$, we observe no significant change in $S_{PCA}$ (we expect a similar behavior for our intrinsic dimension calculations). Therefore, we have kept the plots in the main text unchanged.

(2) We appreciate the referee for pointing this out to us. In Fig. 5a, we can observe that for smaller system sizes the numerical derivative doesn't have a prominent peak, affecting the precision on the estimated $T^{\ast}(L)$ (which is the position of the local maximum in the derivative of the entropy). For larger system sizes the peak is more pronounced and therefore one can locate $T^{\ast}(L)$ more precisely. Further, we observe that the interpolation is more affected by fluctuations in the sampling for the smaller system size as compared to larger lattices. We have amended the manuscript to make this point clear.

(3) We thank the referee for raising this point, as it was probably not so clear in our original manuscript. The interpolation error is accounted for in the following way: We generate a smooth spline approximation for each of $N_b=10$ batches of data (generated as described in the main text) and identify the associated position of the local maximum $T^*_i$ of each of the spline curves. Afterward, we calculate the average $\overline{T^*}$ and estimate the standard error according to the subsampling algorithm described in Appendix B. We have added a sentence to our manuscript to clarify this point.

---

## Round 3 · Referee Report · Biagio Lucini · 2023-11-13

Strengths
(1) The paper provides a robust analysis of phase transitions using two different unsupervised machine learning methods.
(2) The paper contrasts strengths and weaknesses of the two approaches in a sound way.
(3) While the paper uses the Ising model as a test system, conclusions seem generalisable, with the results possibly opening new avenues of investigation in Statistical Mechanics and Quantum Field Theory at finite temperature.
Weaknesses
In my opinion, the paper does not have any weaknesses.
Report
The authors have addressed all the points I have raised in my first report. As the paper easily meets the criteria for acceptance of this journal and is highly relevant, I recommend acceptance of the manuscript in its current form.
Requested changes
N/A

---

## Round 3 · Author Response

We thank you for handling our manuscript and for communicating to us the Referee report. We are also grateful to the Referee for their time in carefully reading our manuscript and providing valuable feedback. We find all points raised by the Referee valid and accurate. We believe that, after addressing the requested changes, our work has been improved.
Below, we provide a summary of the changes made in our revised manuscript.

---

## Round 3 · List of Changes

1. A discussion about the effect of increasing the number of batches in the computation of our estimates has been added in Appendix B.
2. An additional discussion has been added in Sec. 4, explaining the size of the error bars in Fig. 5b, according to the reply to the Referee question below.
3. A short explanatory discussion has been added in Sec. 4 on accounting for the interpolation error and estimating the error bars in $T^*$.

---

## Editorial Decision

published